# Experimental Studies of Welded Joints in Structures Subject to High Impact Vibrations Using Destructive and Non-Destructive Methods

**DOI:** 10.3390/ma16051886

**Published:** 2023-02-24

**Authors:** Piotr Wróblewski, Mariusz Niekurzak, Stanisław Kachel

**Affiliations:** 1Faculty of Mechatronics, Armament and Aerospace, Military University of Technology, ul. Gen. Sylwestra Kaliskiego 2, 00-908 Warsaw, Poland; 2Faculty of Engineering, University of Technology and Economics H. Chodkowska in Warsaw, Jutrzenki 135, 02-231 Warsaw, Poland; 3Faculty of Management, AGH University of Science and Technology, 30-067 Krakow, Poland

**Keywords:** welding, rails, strength analysis, spark butt welding

## Abstract

This article presents the issues of control and quality assurance of high-strength railway rail joints. Selected test results and requirements for rail joints made by welding with stationary welders on the basis of the requirements included in the PN-EN standards have been described. In addition, destructive and non-destructive tests of weld quality were performed, including visual tests, geometrical measurements of irregularities, magnetic particle and penetration tests, fracture tests, observations of micro- and macro-structure and hardness measurements. The scope of these studies included conducting tests, monitoring the process and evaluating the results obtained. Laboratory tests on the rail joints confirmed the good quality of the rail joints from the welding shop. Increasingly less damage to the track in places of new welded joints is proof that the methodology of laboratory qualification tests is correct and fulfils its task. The presented research will help educate engineers on the welding mechanism and the importance of quality control of rail joints during their design. The results of this study are of key importance for public safety and will improve knowledge on the correct implementation of the rail joint and how to conduct quality control tests in accordance with the requirements of the currently applicable standards. It will help engineers choose the right welding technique and choose solutions to minimize cracks.

## 1. Introduction

Along with the development of technology, the demand for diagnostic tests of structural elements is growing. This situation has resulted in the development of these diagnostic tests, allowing to obtain more information about the tested element and to increase the effectiveness of the test. In the operation of the railway infrastructure, an important element influencing the development is the responsibility for the quality of the components of the turnout [1,2]. This applies to the safety of thousands of people traveling by trains, but also of the people who operate them. An undetected defect at the production stage can generate very high costs at the operation stage.

Rail end connections are an important element of the track related to the safety of trains. Joints occur on straight sections and curves of the track, as well as at junctions of turnouts and crossings. Technological difficulties occurring during the execution of rail joints make them vulnerable to local defects and faults. Due to security, the high quality of the connections is very important. This quality should comply with the applicable standards: PN EN 14587-1:2007 [3] and PN EN 14587-2:2009 [4]. The most commonly used method of joining rails is welding [5]. This method requires great care and accuracy in the preparation of the rail ends, compliance with the heating and welding parameters, precise setting and sealing of the form, etc. For this reason, in order to ensure the appropriate quality and repeatability of the process, it is necessary to precisely and repeatably control the process parameters and constantly monitor, control, analyze and evaluate them. In addition, parameters related to post-weld heat treatment and flash removal should be taken into account.

Unfortunately, we have no information on the exact reasons for the lack of standardization of rail link control techniques on a global scale. It is possible that this is due to the lack of uniform standards and methods of measurement on a global scale, as well as the lack of sufficient data and research on the effectiveness of various control techniques. Until it is clearly proven which techniques are most effective, it will be difficult to standardize them.

With these technological aspects in mind, the aim of this article is to construct a test joint of rails in the laboratory by welding and to control the quality of the joint using destructive and non-destructive testing methods. Current regulations were taken into account to assess the implementation of this objective in Poland and the technical and technological parameters of the analyzed devices and operating conditions. The content of the article gives a fresh and innovative look at the analyzed aspects of research. In addition, the article provides the necessary knowledge about the benefits of using the technology described in the work and why this technology can be an important element in improving the quality of railway joints. The rest of the article is presented as follows: Section 2 presents an overview of the global and national literature on the welding of railway joints. Section 3 describes the research methodology used to analyze the research problem. Section 4 describes the results and discusses them. Finally, Section 5 describes the conclusions of the study.

## 2. The Essence of the Research Problem Based on the Literature

Railroad tracks are subjected to tremendous stresses that affect their structural integrity and can cause rail breaks. These phenomena are a serious problem mainly for railway management bodies. The result of these defects can be derailments and, consequently, material losses and damage to the rolling stock. Therefore, all procedures and activities related to track maintenance and inspection have a huge impact on rail traffic safety. In recent years, various technologies have been studied to monitor rails and detect their defects. However, despite significant technological advances in this field, significant research efforts are still required to achieve higher scanning speeds and improve the reliability of diagnostic procedures.

It is well known that rails are subject to intense bending and shear stresses, plastic deformation and wear, which determines the progressive degradation of their structural integrity. Joining railway rails by welding methods creates many technical difficulties; therefore, the number of joining methods that can be used in series is limited. Technical and technological difficulties in joining rails by welding methods are mainly conditioned by the following parameters: the chemical composition of the rail steel, the section and profile size, the high requirements for railway connections and the requirements of high repeatability of welding parameters. Joining railway rails is limited to three methods: electric spark butt welding, thermite welding and electric arc welding [6] (Figure 1).

Additionally, the technical conditions to be met by railway facilities and their location take into account three joining methods: electrofusion, thermite or arc welding. In practice, the electric arc welding method is used sporadically; most often to make a temporary joint or in hard-to-reach places. Technical and technological limitations and difficulties result mainly from the following reasons: the chemical composition of steels used for the production of rails, the cross-sectional size and profile of rails, the high requirements for glued joints of connecting rails and the requirement for high repeatability of welding process parameters, which guarantees repeatability of joint properties [7]. Additionally, the complex cross-sectional profile of rails of different thicknesses is a factor that makes rail welding difficult due to the formation of a very unfavorable distribution of heat introduced into the rail material during arc welding or gas welding. As a result, a very wide heat impact zone is created in the vicinity of the weld, which has an unfavorably changed structure and consequently changed mechanical properties, especially plasticity indices of elongation, impact strength and hardness [8]. Additional factors when choosing the gluing method are the size and shape of the splices on the joint and the possibility of their easy removal and disassembly in order to bring field connections to the rail profile [9]. Bearing these aspects in mind, in accordance with Polish technical standards, the connection of rails in the track should be made by welding with track welders in accordance with the technical conditions Id-1 and the PN-EN 14587-2 standard. However, it is allowed to use thermite welding of rails in places inaccessible to welding electrodes, in accordance with instruction Id-5 [10]. This national management approach to using the preferred method of joining rails in track is due to the fact that cracks in thermite joints were very common in the past. The non-contact track is built of long rails that are welded in stationary welding machines. Thanks to these technologies, a track is obtained in which there is no gap between the rails, as this is the case with the traditional and archaic bolt connection [11,12]. This type of connection can be found on old-type track sections. The gap between the rails is a source of noise in the form of the clatter of wheels of passing rolling stock. Welding technology uses spark resistance welding, which consists of heating the ends (contacts) of adjacent rails by sparking liquid metal in the contact area as a result of current flow, and then applying an upsetting pressure. Welding of rails in the track is carried out using mobile machines moving only on rails or two-way vehicles [13,14]. The welding process should be carried out with automatic machines with a programmed process line, in accordance with the PN-EN 14587-2 standard. The welder should display and record information regarding the splicing program and identifying the rail joint and setting details, such as amperage, swelling force or pressure, rail offset (shortening) and welding time. The process of welding includes individual phases, also known as stages: preparation of rails for welding (appropriate close-up of the front surfaces of the rails), initial sparking level the front surfaces, heating, correct sparking, sparking with increasing speed, upsetting under current characterized by a high rate of shortening, annoying, flash trimming and thermal treatment [15]. The welding program does not have to include all phases, and the rails can be welded with various parameters of current intensity, swelling time and pressure [16]. The culmination of the welding process is obtaining a connection of two rails with a cut tab, which still needs to be ground.

Many researchers, both domestic and foreign, have investigated the subject of this article in their works. Due to the length of the article, the most important works and conclusions of selected researchers are presented. Kaewunruen et al. [17] studied the influence of dynamic loading conditions caused by train–track interactions. The results of the research are recommendations for engineers regarding the adjustment of the support profile and the development of solutions for mitigating excessive carrier/ballast. The authors in paper [18] presented a fatigue analysis of a steel joint using simulation programs. The results of their research were the precise determination of fatigue crack propagation locations and confirmation of the legitimacy of using numerical models to locate cracks. The fatigue assessment of the welded joints between the deck and the rib under long-term load was carried out by the authors of paper [19]. In turn, the authors of the work [20] examined the increase in the mechanical properties of railway frogs made by welding details of Hadfield steel and rail steel by forming and surfacing intermediate alloy inserts. The authors of the next paper [21] studied the possibilities of using IRSM41-97 rail steel in the railway industry. Murakami et al. [22] in turn studied the possibilities of evolution of fatigue damage and fatigue life welded rail connections. In turn, the authors of [23] studied the problem of pattern recognition of structural elements of railway rails on defectograms of multi-channel eddy current flaw detectors using a neural network based on the open Tensor Flow library.

Many researchers have also evaluated various simulation methods and experimental studies in the field of non-destructive and destructive testing in the evaluation of similar solutions used in transport. Such detailed analyses are presented in selected papers [24,25,26,27,28,29]. The impact of particulate matter on the assessment of the technical condition of vehicles and infrastructure is also significant [30,31,32,33,34].

In turn, the authors of paper [35] studied the contact–impact forces in the vicinity of cracked rails, using a detailed three-dimensional finite element model. This model was verified by field tests carried out on track ballasts. In turn, the authors of [36] conducted experimental research using a probe search engine for detecting defects in railway rails. The tests were aimed at structural registration of track damage elements using magnetic field dispersion. Many other authors [37,38,39,40,41,42,43,44,45,46,47], taking into account the latest foreign literature, have investigated problems related to railway connections. Their research was focused mainly on the microstructure of joints, welding methods, numerical methods of crack detection, selection of materials for rails, rail bed conditions, etc.

Based on the review of the literature, it can be concluded that no author has examined the quality of a rail joint made on the basis of destructive and non-destructive methods from a laboratory point of view. This article discusses the economic, technical and regulatory aspects of rail connections made by welding. When assessing the quality of the weld, the climatic conditions, the technical specification and the social, legal and normative requirements were taken into account, which is an innovative approach to the analysis of this problem. The obtained results will allow engineers to make a rational decision regarding the possibility of effective implementation of the railway connection with full knowledge in this regard. The innovativeness of the work consists of the detailed development of measurement methods used in the assessment of welded joints in rail transport. The results of this research give an answer to which methods are worth using from an economic point of view, in particular in terms of the quality of the assessment of these links. The tests are based on an in-depth analysis of welds for unusual structures with high impact loads. A summary of the various causes of rail failure is given in Table 1.

## 3. Materials and Methods

### 3.1. Execution of the Rail Joint

Welding is a method of joining materials consisting of creating a contact surface of joined elements of common grains, resulting from the diffusion and recrystallization of adjacent metal grains. Pressure, temperature and duration of welding are mainly responsible for the effectiveness and correctness of the welding process.

Currently, in Poland, two types of rails, type 49E1 and 60E1, are used on standard-gauge lines. UCI60 and S49 rails are produced in continuous operation, they differ from the currently used rails in details in the transverse profile. In Poland, rails are produced from two types of carbon–manganese rail steel, i.e., R260 and R350HT. The R260 rail steel grade is considered to be a soft rail; the hardness scale is 260–300 HBW, read on the Brinell scale. Its minimum tensile strength is 880 MPa. Rail steel R350HT is considered a hard rail; the hardness is in the range of 350–390 HBW. The hard rail has a much higher tensile strength of 1175 Mpa [48,49].

When making the rail joint, which was the subject of research and analysis, the spark butt welding method was used, which is faster and more economical compared to welding.

Carbon–manganese steel R260 and R350HT are relatively strong steels that are often used in the manufacture of railway rails.

Below are some of their basic parameters:Steel R260:
-Yield strength (ReH): 260 N/mm^2^;-Strength limit (Rm): 380–420 N/mm^2^;-Elongation (A5): 26%;-Modulus of elasticity €: 200,000 N/mm^2^.Steel R350HT:
-Yield strength (ReH): 350 N/mm^2^;-Strength limit (Rm): 550 N/mm^2^;-Elongation (A5): 16%;-Modulus of elasticity €: 210,000 N/mm^2^.

The chemical composition, Vickers hardness values and lamellar spacing of R350HT and R260 rail steels are shown in Table 2 and Table 3, respectively.

To construct the joint of the high-strength rails R260 and R350HT in the laboratory, spark butt welding technology was used with K922 machines. This technology is based on the spark pulse welding method. On this basis, the optimal mode of pulse welding, previously determined by the program, and the temperature distribution in the heat-affected zone were determined. In this heating mode, welding takes 70–80 s. This means that it is reduced three times compared to the canonical mode applied to non-tempered rails. The average melt energy doubles, but the maximum short-term energy consumption remains the same as in discontinuous welding, so when using pulse welding, there is no need for energy sources from the rail welding machines. This technology was chosen because in order to obtain the required mechanical properties of the welded joints, it is necessary to reduce the energy input compared to existing technologies for rail joints not subjected to heating. The new generation K922 machines for welding rails are additionally equipped with automatic multi-factor spark control systems that ensure accurate reproduction of previously set technologies for welding high-strength rails during the repair and construction of railway tracks. In addition, the use of a K922 machine allows the use of a new technology to stabilize the states of temperature stress on continuously welded tracks using stress welding.

The K922 machine is a specialized device used to stabilize the states of temperature stresses on continuously welded tracks. The implementation of this technology allows to improve the quality and safety of railway tracks. Stress welding is one of the latest technologies used in the railway industry. This process involves the use of a special device that heats a piece of track to a high temperature and then immediately cools it down. This process changes the mechanical properties of the steel, making it more resistant to temperature changes and stress. The K922 is able to control and regulate the stress welding process, resulting in consistent quality and safety throughout the track. The use of this technology in combination with other methods of testing and inspecting railway tracks allows for increasing the durability and safety of railway infrastructure.

In Poland, methods and tools for welding thermally uncured R260 rails, developed by PWI, have been used for many years. In this case, continuous flash butt welding (CF) technology was used.

Figure 2 and Figure 3 show a typical joint subjected by the authors to laboratory welding operations and quality control tests.

In order to make the joint, the ends of the rail, cleaned to a metallic sheen, were inserted into the stationary welder and placed butt-end, with a small clamping force, while maintaining vertical linearity (Figure 4).

The chemical composition was obtained by spectrometry. The interlamellar spacing was obtained by drawing a line perpendicular to the cementite lamellae in each pearlite colony and measuring its length, then dividing it by the number of lamellae captured. The method was repeated on five different samples to obtain an average spacing. To view the resulting splint microstructures under an optical microscope (OM) (Figure 5) and a scanning electron microscope (SEM) (Figure 6), the splint samples were ground and polished to a surface finish of 1 μm and etched with 2% nitalic acid. 

A current was passed through the contacting surfaces causing pulsating melting of the metal on the contact surface of the ends of the rails. This resulted in a rapid ejection of the melted metal by the formation of current bridges due to the action of electromagnetic forces and the pressure of metal vapors. In addition, under the influence of the resulting sparking, the remains of impurities were simultaneously removed from the front surfaces of the joined rails and the ends of the rails quickly heated up due to the penetration of heat into the material. After the sparking process, the upsetting process was initiated with the force causing the rails to be crushed and the material to be flashed at the welding spot. The resulting flash was removed using profile knives at a temperature of 500 °C, and then the joint was ground. During the welding process, the microstructure of the rail joint was shaped by recrystallization, casting, phase crystallization and other thermal and mechanical processes. The contact center rail joint microstructure varies with material type and welding conditions, but can include hardening and leaching areas, areas of phase formation and other microstructural changes. The microstructure of the rail joint for selected materials is shown in Figure 5.

An analysis of the microstructure in the center of the weld and in the heat-affected zone shows that the grain size is 2–3 times smaller and the ferrite level is significantly reduced compared to the values obtained for continuous welding. The structure in the center of the weld and in the adjacent area of the heat affected zone remains as sorbitol-perlite and sorbitol; there are no dangerous hardening structures. The joint made in this way was then subjected to quality control.

### 3.2. Checking the Rail Joints

Railway rail joints are subject to qualification tests, covering the scope of tests recommended by the standards. The tests were performed on ten test joints. Operational tests were also carried out on ten rail joints on a selected track section during one year of joint operation. These tests included control of the joint manufacturing technology, measurement of the step size and weld flash (in the case of a welded joint), measurement of vertical straightness on the rolling surface and horizontal straightness on the rail surface, penetration and ultrasonic tests, a HV30 hardness test, a static bending test and macroscopic, microscopic and fatigue strength tests of the material.

Testing rails after one year of operation is often practiced, but the frequency of testing depends on many factors such as traffic intensity, weather conditions, rail design, etc. Testing after a longer duration, such as 5 or 10 years, is rarely practiced because rails are regularly inspected and reviewed during this period. The typical inspection interval for railway rails depends on many factors and can range from a few months to several years. It all depends on the degree of traffic intensity, climatic conditions, type of rails and other factors. It is necessary to adapt the inspection interval to the specific situation and needs. Therefore, this study assumed that one year would be the optimal value. Such studies are quite often carried out in practice.

The technology control of the welded joint includes cleaning procedures and alignment of the rail ends, setting the current linearity in the welder handle and parameters, sparking time and upsetting force. The correct cutting of the material flash after upsetting affects the labor consumption of the process grinding of the rail head, geometry and quality of the welded joint.

The accuracy of setting the ends of the rails and the size of the flash left and the riser after welding are the elements that determine the obtained parameters of vertical and horizontal straightness of the rail joints, as well as the labor consumption associated with the grinding operation of the rail joints. The requirements for these parameters are listed in Table 4 and Table 5.

The tests on the welded rails were carried out in accordance with the requirements of the European standard EN 14587-1-2007. The welded contacts of all series were monitored by non-destructive testing (ultrasonic and capillary methods). They have also been tested for static mechanical bending in accordance with the requirements of this standard.

Among the methods of diagnostics of welded joints, non-destructive testing is used. Non-destructive testing (NDT) is a research method that provides information about the properties of the material without affecting its structural properties. The purpose of the test is to detect and evaluate material discontinuity defects at the initial stage of production. The later a defect is detected, the more costs are generated by this failure. In the case of weld testing, the following non-destructive tests were used:Visual examination (VT);Penetration testing (PT);Magnetic particle testing (MT).

Visual research is based on careful inspection of the surface. An endoscope with a prism was used for this, which allows one to change the angle of view of the image. This is a basic test used for all kinds of joints, forgings, castings, pipes of various diameters and plastically processed products. In order for the test to be conducted successfully, the following conditions must be met:The illuminance on the test surface should be at least 350 lux, but the recommended illuminance value is 500 lux;The distance between the tested object and the examiner’s eye should not exceed 600 mm;The viewing angle must not be less than 30°;The tested surface must be clean and free of dirt and grease.

Penetrant testing is mainly used to detect discontinuities in metal elements, e.g., welds, castings or forged elements. The penetration method allows to detect inconsistencies that occur on the surface, e.g., cracks, pores and sticking. This method is also applicable in the case of non-magnetic materials, e.g., aluminum, titanium and austenitic steels. The test has a wider range of applications compared to the magnetic test, which is limited to ferromagnetic materials. With polished surfaces, very small material discontinuities with an opening width of 0.5 µm to 1 µm at a depth of 10 µm can be detected without any problems. With the increase in the roughness of the tested surface, the sensitivity of detecting defects decreases. Surface roughness is described by R_a_, but in practice it is difficult to determine a surface suitable for testing. The face of the welded joint has characteristic scales, and the transition of the weld into the base material is also difficult to assess. It is the duty of the personnel supervising the examination to assess whether the presented surface is able to be tested or whether it requires appropriate preparation. Standards are recommended to determine the R_a_ value. If the surface roughness is too high, it is recommended to treat it mechanically. The method of treatment should be chosen so as to avoid a negative impact on the test. When grinding the surface, this should be performed perpendicular to the expected direction of the discontinuity to avoid contamination of the gap by filings. It should also be noted that when using tools, e.g., a chisel, too much pressure on the surface may cause the gap to close. The limitation of this method is the detection of surface defects; discontinuities located below the surface will not be detected. The downside is that proper preparation of the surface for testing requires a lot of cleaning, and the test itself is also not fast because the penetrant needs time to absorb into the surface.

The method of magnetic particle testing belongs to the group of non-destructive tests. This type of test is widely used in the aviation, automotive, energy and many other industries, where ferromagnetic materials are used. MT research finds applications where surface defects as well as discontinuities occurring under the surface are found. This technique can be used to test welded joints, castings and forgings. Magnetic particle testing is a quick and safe method of finding discontinuities in a material. Thanks to the high sensitivity, material defects can be found on the surface as well as at a small depth. The advantage of MT testing is the ability to test an object regardless of its shape and size, one just needs to choose the appropriate testing technique, e.g., a method that uses non-fluorescent magnetic powder. The most commonly used magnetic powder is black or reddish-brown. When using black powder on metallic, gray or rough surfaces, the test contrast is low and often leads to false discontinuity readings. A reddish-brown powder can be used to enhance the contrast. When performing a study on smooth surfaces using black powder, the indications are clearly visible, but there are often reflections on the surface that cause glare. In order to increase the contrast of the examination, a primer paint may be used. By using paint, we increase the recognition of discontinuities, but we lose the sensitivity of the test, because the paint increases the distance from the test surface.

## 4. Results and Discussion

### 4.1. Checking the Rail Joints

The testing of railway rails can be carried out using non-destructive or destructive methods. Below is a description of some of them:

Non-destructive methods:Ultrasonic testing: Uses an ultrasonic wave to detect inhomogeneities inside the rail, such as cracks or changes in density. The wave is sent down the bus and received on the other side, which allows any anomalies to be detected. An ultrasonic wave is sent into the rail and received on the other side, which allows any abnormalities to be detected. The mathematical formula that describes the motion of an ultrasonic wave is the equation of wave motion:
Δ = λ/2 * (n_1_ − n_2_)
where Δ is the amplitude of the waves, λ is the wavelength and n_1_ and n_2_ are the refractive indices of the material on both sides of the surface. 

Magnetic-powder test: This consists of applying a special magnetic powder to the surface of the rail and applying a magnetic field. If there are any defects inside the rail, the magnetic powder will concentrate there, allowing them to be detected. The mathematical formula that describes the focusing of magnetic powder is:B = μ * (I * dl)/2 * π * r
where B is the magnetic force, μ is the magnetic permeability, I is the current, dl is the length of the wire and r is the distance from the wire.

Thermographic examination: Measures the temperature of the rail to detect inhomogeneities inside. Any changes in temperature indicate the presence of defects such as cracks or changes in density. In this method, the rail is mechanically loaded and then the surface temperature of the rail is recorded using thermography. Inhomogeneities within the material, such as cracks, lead to temperature changes that are visible in the thermographic image.

This can be described by the following mathematical formula:dT = λ * ∇^2T + Q
where dT is the temperature change, λ is the thermal conductivity of the material, ∇^2T is the Laplace temperature variance operator and Q is the source of internal heat in the material.

This formula describes how heat is displaced within the material, allowing inhomogeneities and defects within the material to be detected by surface temperature measurements.

Destructive methods:Strength test: This consists of subjecting the rail sample to an appropriate load to determine its strength. After the sample has been destroyed, microscopic analysis can be performed to determine the cause of the damage. Strength testing of railway rails is one of the most important tests that are used to assess the strength and resistance of rails to damage and cracks during operation. In this test, the rail is subjected to a mechanical load and then the force required to break it is measured. One commonly used approach to strength testing is the constant load approach, where the rail is subjected to a constant mechanical load and then the force needed for it to fail is measured. This can be described by the following mathematical formula:
σ = F/A
where σ is the stress, F is the load force and A is the cross-sectional area of the rail.

Another approach to strength testing is the length change approach, where the length of the rail is measured as it is subjected to mechanical stress. This can be described by the following mathematical formula:ΔL = ε * L
where ΔL is the change in length, ε is the deformation factor and L is the length of the rail.

These mathematical formulas are used in conjunction with mechanical measurements to determine the strength of rails and their ability to withstand loads in service.

Chemical Composition Testing: This involves taking a sample of the rail and performing a chemical analysis to determine its composition and quality. This method allows the detection of irregularities related to the production of rails, such as inadequate carbon content or insufficient content of other components. Testing the chemical composition of rails is one of the most important tests that is used to assess the quality and durability of rails. The purpose of this test is to determine the chemical composition of the material, including the amount of individual elements and the degree of its homogeneity. The procedure for testing the chemical composition of rails with regard to defects and faults may vary depending on the method and technology used by the laboratory, but generally consists of several steps:
-Sample preparation: the rail sample is cut and subjected to preparation processes such as milling, shot peening or grinding to obtain the sample surface for analysis.-Chemical Analysis: chemical analysis is performed by various methods such as mass spectrometry, atomic absorption or emission spectrometry to determine the chemical composition of a sample.-Quality Assessment: chemical analysis is used to assess the quality of the splint, including its degree of homogeneity and the presence of any undesirable impurities or defects.-Fault Identification: if faults or defects are detected, further analysis is performed to determine their type and cause.-Reporting of results: at the end of the procedure, the results of the chemical composition test are reported in the form of an official document, which includes information on the chemical composition, quality and presence of faults or defects.

This process is important because it allows the durability and strength of the rails to be assessed, which is essential for the safety and reliability of the rail system.

Non-destructive testing of railway rails has the following advantages and disadvantages.

Advantages:-It does not damage the rails, which allows them to be reused and reduces disposal costs;-The speed and reliability of the test increases the safety of railway traffic;-They can be performed spontaneously, which allows for quick detection of faults and defects.

Disadvantages:-The accuracy of the results may be lower than those of destructive testing;-It requires special equipment and trained operators, which increases the cost of the examination;-It does not give a complete picture of the chemical composition and strength of the rail.

Destructive testing of railway rails has the following advantages and disadvantages.

Advantages:-It gives a full picture of the chemical composition and strength of the rail material, which allows one to accurately determine its quality and durability;-It is very accurate and its results are easily proven;-It allows the detection of defects and faults that could pose a threat to the safety of railway traffic.

Disadvantages:-The test is destructive, which means that the rail is damaged or destroyed during the test;-Several tests may be required to get a full picture of the rail chemistry and strength, adding to the cost of testing;-Sometimes it is difficult to determine whether a fault is serious or not, which can lead to unnecessary rail replacement.

Laboratory tests of welded rail joints were carried out on ten sections of rail joints made with Schlatter welding machines. The results of the straightness tests carried out on the running surface of the head of the welded rail joints show that the weld sections of the rails have a vertical convexity and concavity f not exceeding 0.20 mm and horizontal f up to 0.25 mm, which proves that the rail connection process is properly conducted, in accordance with the requirement standards. The horizontal straightness of the weld measured on the edge of the rail foot did not exceed 1.50 mm and was within the tolerance requirements, which also proves that the rail connection process was properly conducted. A 1 m-long ruler was used to check the straightness (Figure 7).

### 4.2. Non-Destructive Testing Program

The subject of the visual examination was the weld (Figure 8) taking into account the HAZ (heat affected zone) 10 mm on each side. The weld for the test was previously prepared, i.e., subjected to mechanical and deformation treatment, and all impurities were removed from the joint area.

The joint surface was examined at 500 lx illumination, while the observations were made at an angle of 30° and a distance of 600 mm. The VT study was conducted in two general and specific stages. In the general examination, the object was identified and the completeness, correctness and folds of the weld were checked. However, in a more detailed study, the geometry, straightness deviations and mutual positions were checked. A ruler, which was placed on the head, was for the checks, and the straightness of the weld was measured using a feeler gauge. After checking the geometry, the search for external and internal discrepancies began using penetration methods. All discrepancies that were detected were measured and checked to see whether they meet the required level contained in the EN ISO 5817/EN ISO 10042 standard. The MR 68C penetrator was used for the test. Penetrant MR 68C is characterized by high sensitivity for detecting material discontinuities, which makes it suitable for use in color and fluorescence techniques. Is based on mineral oil and contains a low content of sulfur and halogens. These characteristics make it suitable to obtain reliable test results. It was applied to the test sample by spraying in accordance with the PN EN ISO 3452-1 standard. The penetration time was 45 min at 25 °C. Figure 9 shows the process of penetration of the penetrant.

After 45 min, the excess penetrant was removed from the tested surface. The solvent cleaning method was used to test the weld. Then, an aerosol developer was used (Figure 10). The use of an aerosol allowed the developer to dry quickly in 30 min through the use of solvents.

The results of the penetration tests are shown in Figure 11, where a lack of surface defects in the form of cracks and discontinuities in the joint area can be observed.

Fatigue resistance testing of butt-welded rail joints is an effective method to confirm the correctness of the butt-welding process. Defects occurring during production affect the strength and plasticity of the joints, which leads to their cracking. These cracks are visible at various times during dynamic fatigue testing. The presence of larger matte surfaces causes cracking of the welds after about 1 million fatigue cycles, while small defects form over 3 million cycles. These imperfections are the source of fatigue cracks in rail joints. Figure 12 shows the cracks in butt-welded joints that have undergone separation cracking during fatigue testing. The internal stresses in the rails and rail joints are also critical for their continued service. Stresses above 250 MPa in rails and rail joints can lead to fatigue cracks during track operation.

In order to detect both surface and subsurface defects in the rail joint, magnetic particle tests were performed. Before starting the test, as in the penetration method, a cleaned joint was prepared for testing. Subsequently, initial magnetization of the surface was carried out (Figure 13) then the application of the suspension was initiated. Magnetic primer MR 76S, which is a black oil suspension, was used as the suspension. The color of the magnetic base is contrasting to the white base, thanks to which the indications appearing at the examination site were easier to observe. Thanks to the use of magnetic powder in the form of an aerosol, the layer was easily applied during magnetization. In the next step, the surface was observed and the indications, both those that were recordable and those that were unacceptable, were recorded. In order to obtain reliable results, the test was repeated using the same steps after shifting the field by 90°.

This test, as well as the penetration test, did not reveal any internal defects in the welded joints. The tested samples are characterized by high strength and plasticity. The R260 rail series did not break during testing, and the loads were at the base metal values. Therefore, an incision was made in the center of the weld and samples were brought to the fracture to check for defects along the joint line. In the R350 rail sample series, 100% of the samples had cracks at the welds, and the loads and bending deviation far exceeded the established requirements. No defects were noticed in the cracks of the samples. All samples of the reference series tested were free of defects after capillary and ultrasonic testing.

A HV30 hardness test of the joints was carried out in accordance with the PN-EN 14587-1:2007 and PN-EN 14730-1:2010 standards. Measurements were made at a depth of 5 mm from the running surface of the rail head. In the R350HT rails, the hardness decreases in specific areas of the zone over a length of 1–2 mm, where the heating has reached the level of steel hardening. In the middle, however, due to the reduction in carbon content in the metal layers adjacent to the welded joint subjected to heating with temperatures close to the melting of steel, the structure of hardened sorbitol with high plastic properties was observed in areas with reduced hardness. The hardness in the softening zone of the joints was within the range provided by the applicable standards, i.e., max ≤ P + 60 HV30 and min ≥ P – 30 HV30, where P is the hardness of the base material of the rail. The tests carried out show that when welding high-strength R260 and R350HT rails, high values of mechanical properties of welded joints can be obtained by maintaining the basic conditions, i.e., a significant reduction in energy consumption compared to technologies acceptable for welding uncured rails.

A bending strength test was carried out on a rail with a 60E1 profile and a hardness of R260. The welding sample was previously prepared. The head and foot of the rail were ground and the edges welded rounded (Figure 14). After this operation, the length of the sample is 1200–1250 mm.

The sample prepared in this way was placed in the press in a special basket. This basket is designed to protect against being hit by a splinter of the rail in the event of the specimen breaking during the bending test. The press used for the test is shown in Figure 15.

The bending test was performed on swinging supports. The sample was placed centrally in the middle, on two supports, in such a way that the roller pressed straight on the weld. Figure 16 shows the correct sample positioning.

With the increase in the force to which the weld was subjected, the deflection value of the sample increased. For the test to end with a positive result, both values, i.e., force and deflection, had to reach minimum values. A positive result was obtained in all tests, i.e., a joint deflection greater than the required minimum of 20.0 mm was obtained at a bending force of 1600 kN for welded joints. The obtained results are presented in Table 6. They comply with the requirements of the applicable standards. The bending diagram of the performed test is shown in Figure 17.

## 5. Conclusions

Conducting regular and detailed inspections of railway rails is extremely important to ensure the safety of passengers and drivers moving on railway tracks. Rails are exposed to everyday stresses, such as train running and weather conditions, which can lead to their gradual wear and weakening. It is therefore important that the rails are regularly inspected and repaired or replaced if necessary. The tests of the rails should include an assessment of their condition, condition and strength, as well as an assessment of possible hazards such as corrosion, cracks and deformations. If any irregularities are detected, immediate corrective actions should be taken to avoid serious failures and accidents. In addition, rail testing should be conducted before any seasonal change, such as winter, to ensure rails are strong enough and ready for winter conditions. In this way, the safety of passengers and drivers can be ensured and delays and interruptions in train movements can be prevented. Finally, conducting regular inspections of railway rails is also important for economic reasons. By regularly maintaining and repairing the rails, more serious and costly failures can be prevented in the future, which can save money for the railway company.

The issue of quality in the implementation of welding processes when making rail joints is of fundamental importance for minimizing damage to the railway surface, and thus ensuring the safety of passengers. However, the best-designed process does not always guarantee its correct execution. For this reason, it is necessary to control the implementation of process parameters for each performed weld. The effect of the work was to make a weld and conduct quality control tests on it using destructive and non-destructive methods. The spark butt welding technology was selected to make high-strength R260 and R350HT rail joints due to it ensuring good contact properties in accordance with European Union standards. The experimental and control tests allow to obtain good quality rail joints of type 60E1 in class R260. Laboratory tests of welded rail joints, including an analysis of the joint execution technology; visual, penetrant and magnetic particle tests; straightness tests of vertical and horizontal connections; macro- and micro-structure tests of rail connection areas; and bending tests, were carried out in accordance with the applicable PN-EN standards.

As part of the research, it was shown that:-When welding high-strength rails, in order to obtain the required mechanical properties of the welded joints, it is necessary to reduce the energy input compared to existing technologies for rail joints not subjected to heating.-The use of the K922 machine allowed the use of a new technology for stabilization of temperature stresses on tracks welded continuously using the stress welding method. The methods of making the welded rail joint were made in accordance with the recommendations contained in the standards and technical conditions.-Laboratory tests of rail joints confirm the good quality of rail joints, both from the welding shop and those made on the track. Increasingly less damage to the track at the location of new welded joints is proof that the methodology of laboratory tests is correct and fulfils its task. This type of research has its limitations. From the scientific point of view, it should be emphasized that the adopted joint production and quality testing techniques, depending on the location and size of defects, may significantly differ in their selection. Penetration testing can be carried out almost anywhere, but this limits the penetration time. Magnetic particle testing is very fast, but the limitation is the current source needed to magnetize the surface by the device. For these two studies, the choice is up to the researcher.

However, these are the basic methods that must be performed to confirm the quality of the joints in accordance with the requirements of PN-EN standards. However, these data are the starting point for developing new solutions in the fields of automation, control and visualization of the processes being carried out. Further research in this area should include friction stir welding (FSW). This technology can be used wherever welds must be continuous and free from cracks and porosity typical of welds obtained in the welding process. In addition, other important factors influencing the quality of connections should be analyzed, taking into account the requirements contained in global standards and tested in domestic conditions.

## Figures and Tables

**Figure 1 materials-16-01886-f001:**
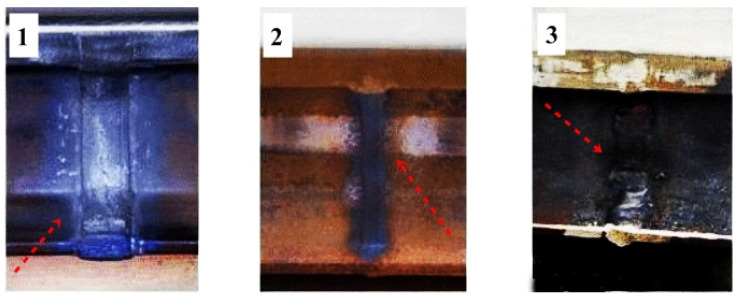
Basic methods of connecting rails, (**1**) thermite welding, (**2**) electric spark welding, (**3**) arc welding.

**Figure 2 materials-16-01886-f002:**
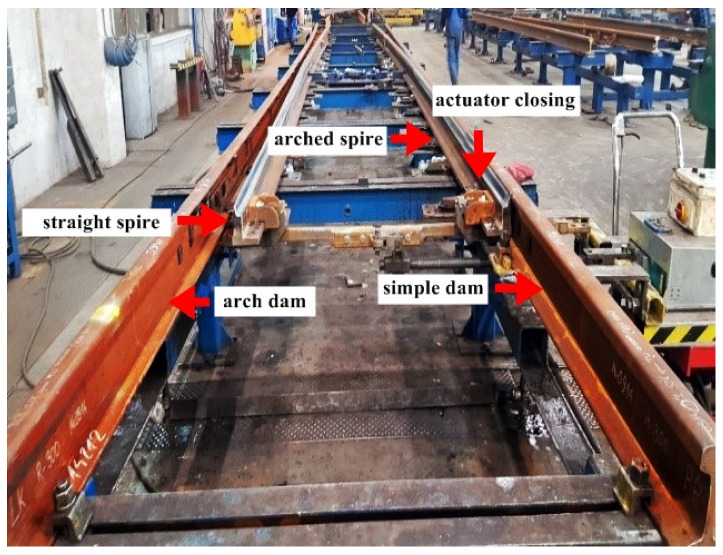
Railway switch. Source: own.

**Figure 3 materials-16-01886-f003:**
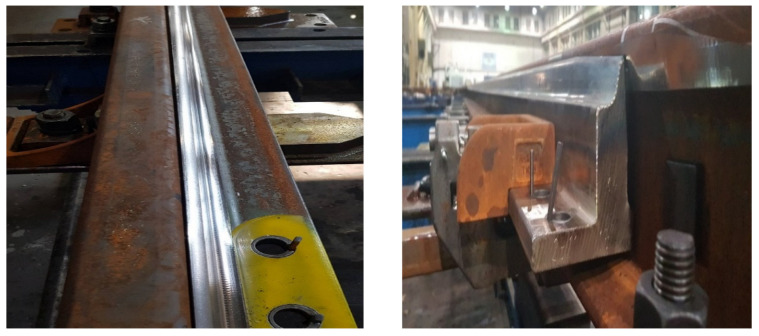
Adhesion of the needle to the stop. Source: own.

**Figure 4 materials-16-01886-f004:**
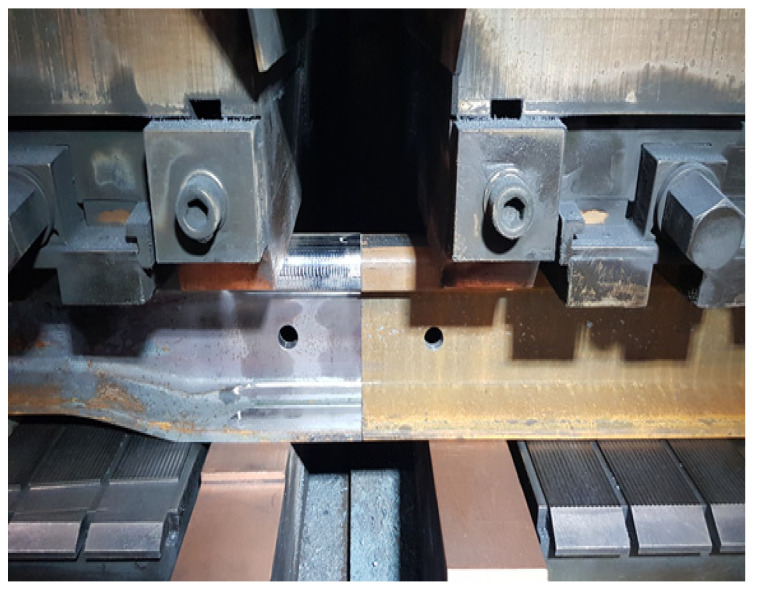
Welding method of rail joints: butt positioning of rail ends in the welder. Source: own.

**Figure 5 materials-16-01886-f005:**
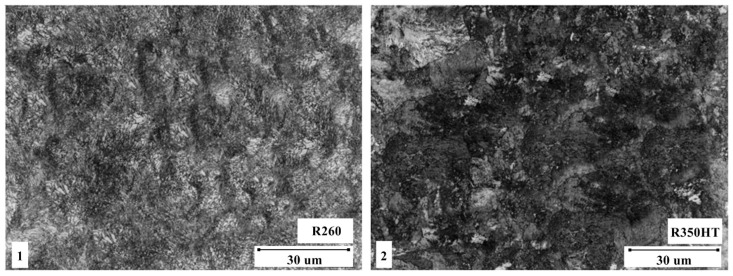
Microphotographs taken with an optical microscope: **1**—a section from the R260 rail, **2**—a section from the R350HT rail, image of pearlitic grains in the microstructure. Source: own.

**Figure 6 materials-16-01886-f006:**
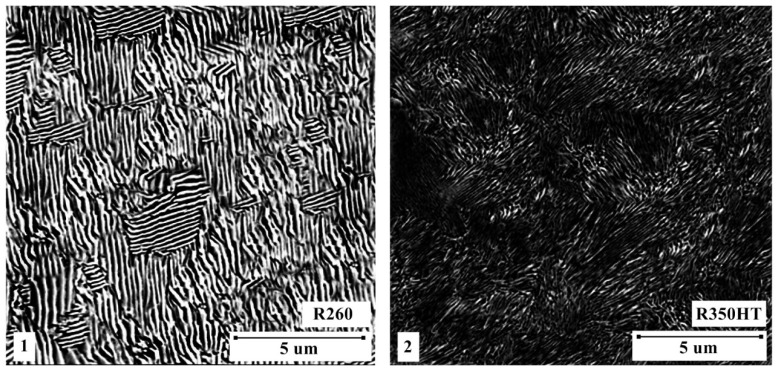
Micrographs taken with a scanning microscope at ×5000 magnification: **1**—samples of the R260 rail, **2**—samples of the R350HT rail, various pearlite colonies composed of alternating layers of ferrite and cementite are visible.

**Figure 7 materials-16-01886-f007:**
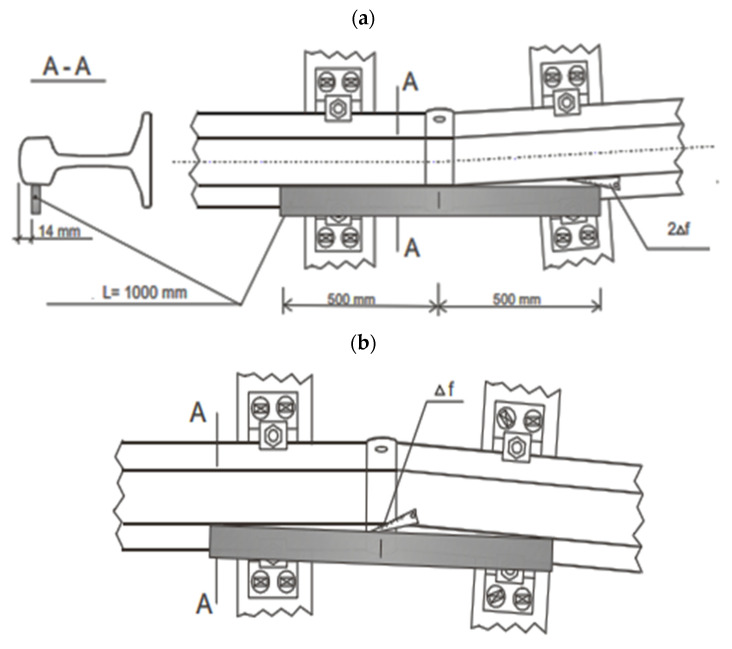
Scheme of checking the straightness of the joint in the horizontal plane: (**a**) convexity, (**b**) concavity. Source: own.

**Figure 8 materials-16-01886-f008:**
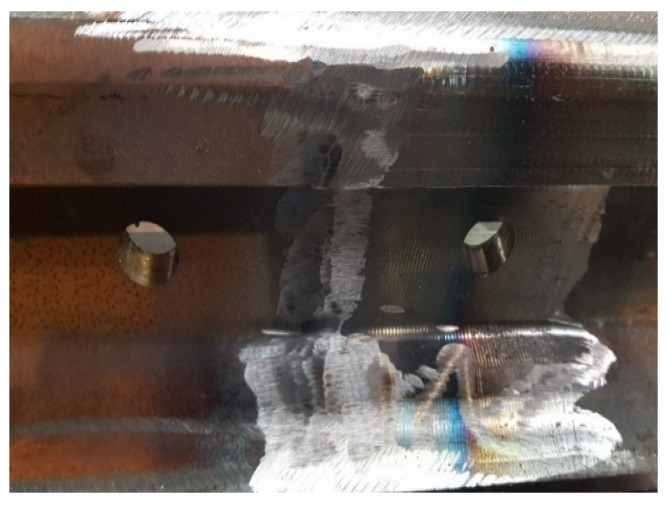
Needle weld. Source: own.

**Figure 9 materials-16-01886-f009:**
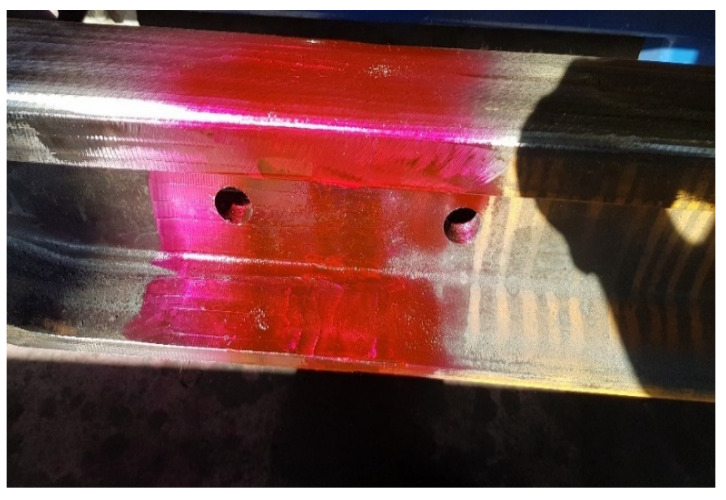
The penetrant penetration process. Source: own.

**Figure 10 materials-16-01886-f010:**
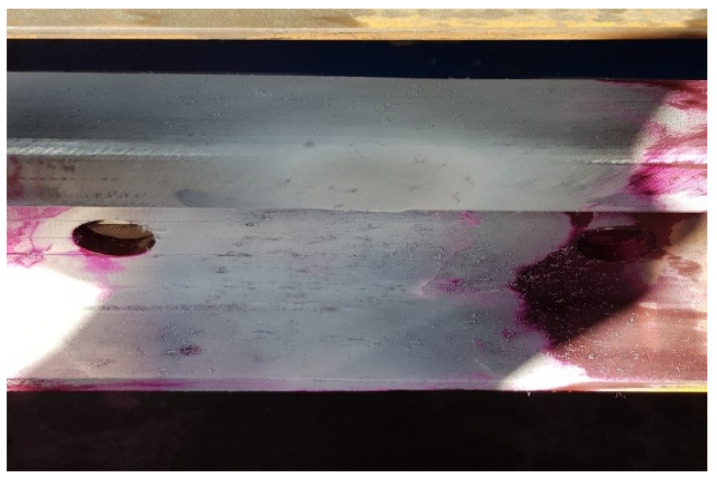
Developer application process. Source: own.

**Figure 11 materials-16-01886-f011:**
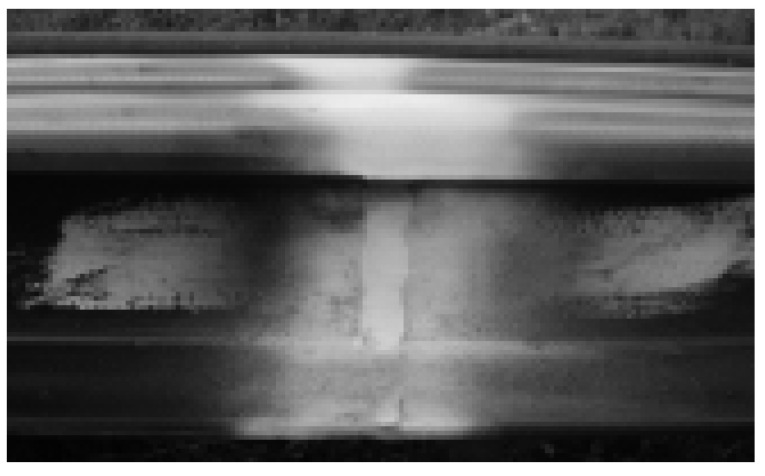
Penetration testing of rail joints. Source: own.

**Figure 12 materials-16-01886-f012:**
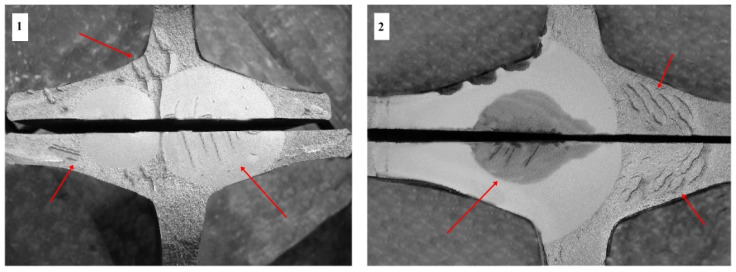
Examples of fatigue fracture of the flash shaft: **1**—welded connection of rails for R260 steel, **2**—butt welded connection for R350HT steel.

**Figure 13 materials-16-01886-f013:**
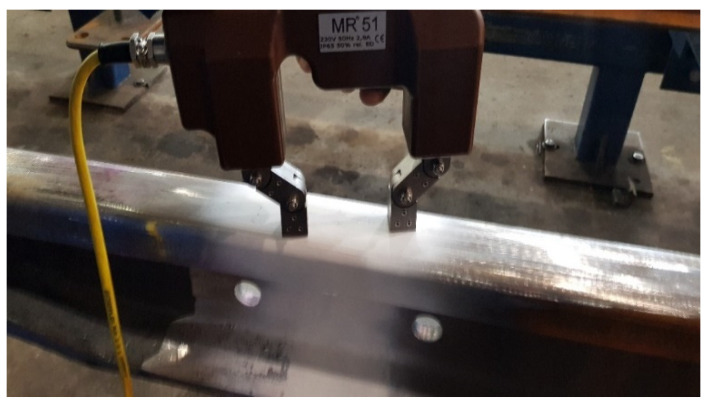
Surface magnetization. Source: own.

**Figure 14 materials-16-01886-f014:**
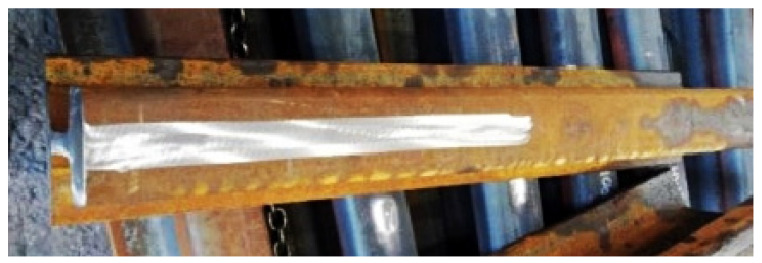
Rail head and foot prepared for welding. Source: own.

**Figure 15 materials-16-01886-f015:**
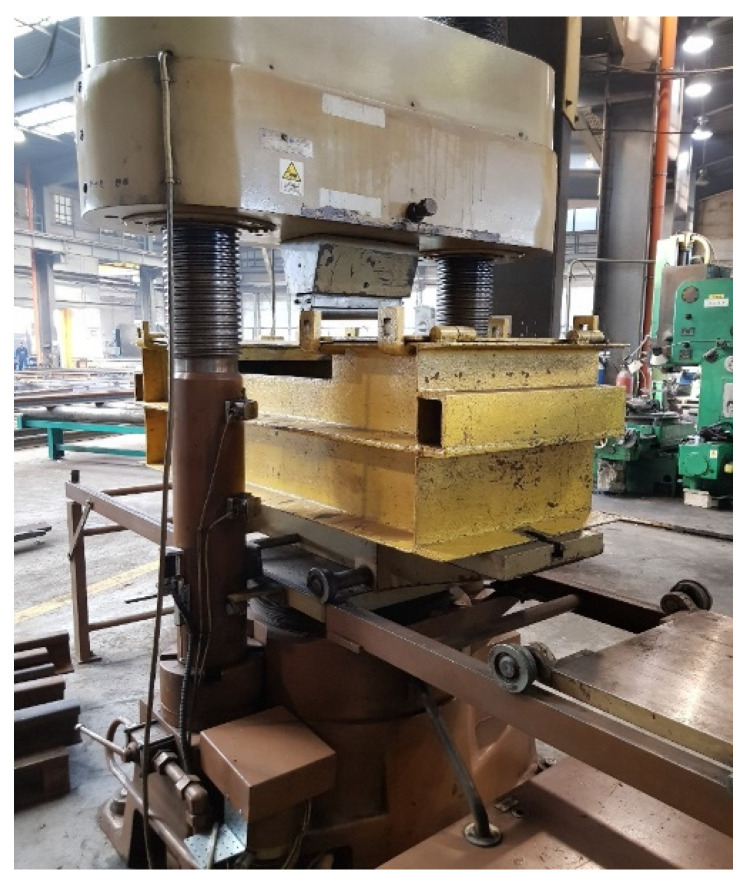
The press used for the test. Source: own.

**Figure 16 materials-16-01886-f016:**
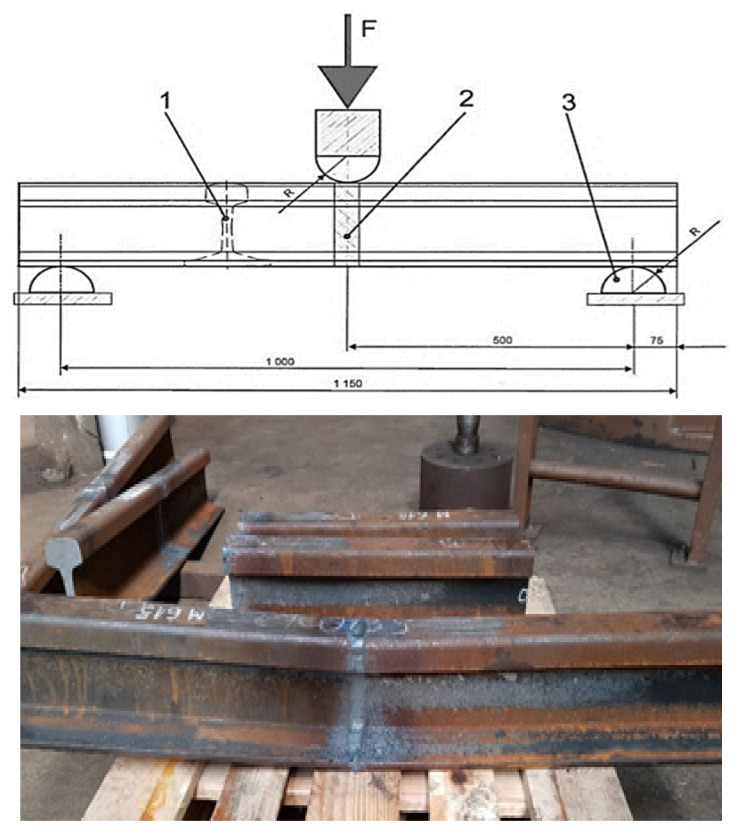
The bending scheme and the appearance of the sample after the test. Source: own.

**Figure 17 materials-16-01886-f017:**
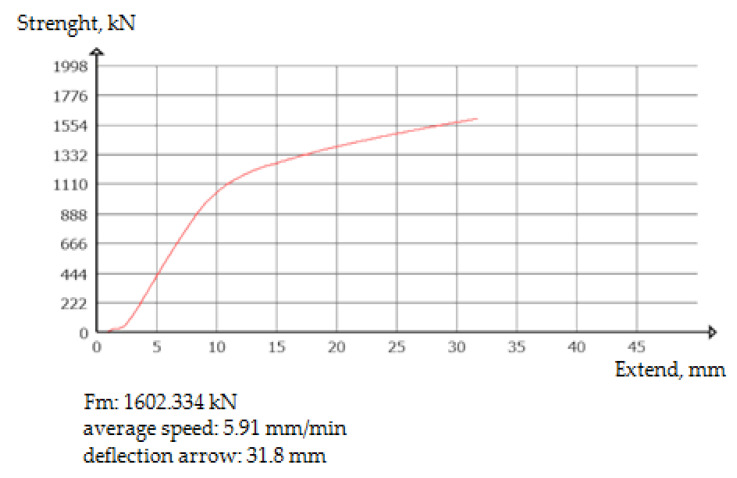
Bending diagram of a welded rail joint. Source: own.

**Table 1 materials-16-01886-t001:** Overview of rail defects and their detection techniques.

Authors	Topic	Reference
Zumpano, G.; Meo, M.A. [48]; Muravev, V.V.; Boyarkin, E.V. [49]; Jeffrey, B.D.; Peterson, M.L.; Gutkowski, R.M. [50]; Abbaszadeh, K.; Rahimian, M.; Toliyat, H.A.; Olson, L.E. [51]	Manufacturing defects in the rails	Damages in this category include transverse defects (TD) and longitudinal defects (LD). They also include inclusions or abnormal local mixing in the rail steel. Progressive cracks developing in the rail face parallel to the transverse direction or an internal progressive crack propagating longitudinally in the rails.
Zumpano, G.; Meo, M.A. [48]	Defects caused by improper use or handling of rails	Damage caused by improper use or handling of the rails is usually caused by the wheels of the train spinning on the rails (wheel burn defect or sudden emergency braking of the train).
Vadillo, E.G.; Tarrago, J.A.; Zubiaurre, G.G.; Duque, C.A. [52]; Bohmer, A.; Klimpel, T. [53]; Cannon, D.F.; Edel, K.O.; Grassie, S.L.; Sawley, K. [54]; Cannon, D.F.; Pradier, H. [55]; Grassie, S.; Nilsson, P.; Bjurstrom, K.; Frick, A.; Hansson, L.G. [56]	Defects caused by wear and fatigue of the rails	Fatigue defects and rail wear result from rolling surface wear and/or fatigue mechanisms. These are corrugation, rolling contact damage or hole cracks. The waviness is related to the wear of the rail head. This increases noise emissions, strain and fatigue. Rolling contact fatigue defects affect structural integrity as they can cause complete rail failure. Recycled fiber damage can be divided into check and possible chipping, flaking and squatting. Liquid absorption by the metal can accelerate the growth of a surface-initiated crack in the rail.
Papaelias, P.; Roberts, M.C.; Davis, C.L. [57]; Ghofrani, F.; Pathak, A.; Mohammadi, R.; Aref, A.; He, Q. [58]	Defective growth	Rail failure caused by cracks in metal structures has three phases: crack initiation, crack growth and subsequent rapid crack growth. Various techniques are used to detect cracks when they are in earlier stages so that the third phase is not reached. The development of defects is determined by uncontrolled and random processes or events; therefore, statistical descriptions cannot be used.

**Table 2 materials-16-01886-t002:** Chemical composition of R260 and R370HT steel.

Steel	C	Mn	P	S	Si	Ni	Cr
R260	0.70	0.91	0.025	0.0049	0.354	0.045	0.036
R350HT	0.79	1.17	0.013	0.016	0.388	0.061	0.197

**Table 3 materials-16-01886-t003:** Vickers hardness and interlayer spacing of rail steels.

Steel	Interlamellar Spacing, nm	Hardness, HV 10
R260	119 ± 10	370 ± 6
R350HT	335 ± 21	295 ± 6

**Table 4 materials-16-01886-t004:** Permissible deviations of horizontal straightness.

Scale Classification	Purpose of the Rail/Dimension Deviations ∆f, mm			
	Main Tracks		Other Tracks	
	Concavity	Convexity	Concavity	Convexity
Okay	∆f ≤ 0.3	∆f = 0.0	∆f ≤ 0.5	∆f ≤ 0.5
For repair	0.3 < ∆f ≤ 0.6	0.0 < ∆f ≤ 0.3	0.5 < ∆f ≤ 0.8	0.5 < ∆f ≤ 0.8
To be cut out	∆f > 0.6	∆f > 0.3	∆f > 0.8	∆f > 0.8

Source: Own, based on [59,60].

**Table 5 materials-16-01886-t005:** Permissible vertical straightness deviations.

Scale Classification	Purpose of the Rail/Dimension Deviations ∆f, mm					
	Main Tracks				Other Tracks	
	V > 160 km/h		V ≤ 160 km/h			
	Concavity	Convexity	Concavity	Convexity	Concavity	Convexity
Okay	∆f = 0.0	∆f ≤ 0.3	∆f ≤ 0.1	∆f ≤ 0.3	∆f ≤ 0.5	∆f ≤ 0.5
For repair	0.0 < ∆f ≤ 0.2	0.3 < ∆f ≤ 0.5	0.1 < ∆f ≤ 0.3	0.3 < ∆f ≤ 0.5	0.5 < ∆f ≤ 0.8	0.5 < ∆f ≤ 1.0
To be cut out	∆f > 0.2	∆f > 0.5	∆f > 0.3	∆f > 0.5	∆f > 0.8	∆f > 1.0

Source: Own, based on [59,60].

**Table 6 materials-16-01886-t006:** Obtained bending test results of welded rail joints.

Type of Rail Joint	Bending Force, kN		Displacement, mm	
	Requirements	Findings	Requirements	Findings
Welded joint according to PN-EN 14587-1:2007	>1600	1638	>20.0	31
		1695		26
		1752		38
		1746		28
		1789		29

Source: own.

## Data Availability

Not applicable.

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
