# Peer review of "Experimental Studies of Welded Joints in Structures Subject to High Impact Vibrations Using Destructive and Non-Destructive Methods"

_materials, 2023, doi:10.3390/ma16051886_

Round 1
Reviewer 1 Report
Chapter 2 should be enriched with figures that sketch the three welding methods (electric spark butt welding, thermite welding, electric arc welding) to interrupt the long chapter. A table which summarizes the main findings from the state-of-the-art literature could be also added to facilitate the reader on the understanding of the literature gaps. There are a few references which are not in english and that I could not check. I would propose to reduce the national literature and to refer to international papers where possible.
Paragraph 3.1 should have a table which summarized the process parameters adopted during welding as well as a table for the composition of the selected materials. It is not reported between which materials the joint was carried out (e.g., R260-R350HT, R260-R260, R350HT-R350HT). The transverse geometry of the rails should be also added. Some of these arguments are reported in paragraph 4.1., but they should be moved back to the methodology chapter, because as stated in the manuscript title and aim, the focus of the paper is on the quality control of these joints using destructive and non-destructive methods. A reorganization of the paper in this direction would help the reader on the understanding of this target.
Paragraph 3.2 and 3.3 could be merged because they discuss of related topics.
Paragraph 4.2. should be organized in subparagraphs (one for each control method) because the chapter is long, and the reader needs to be guided through the manuscript. The organization of the paragraph could see first the non-destructive analysis and then the destructive ones. Regarding the destructive methods, a joint macro should be reported with a 5x magnification to provide an overview of the entire butt joint showing Base material-HAZ-Fused Zone-HAZ-Base material (after etching). Then, higher magnifications can be added for the most important areas showing the grain structure and their position on the main macro at 5x. The “HV30 hardness test” is missing.
Finally, the authors should have a paragraph (or at least a table) where they discuss and compare the potentialities (e.g., advantages and disadvantages) of each selected destructive and non-destructive methods to reconnect the results with the aim of the paper. The authors may merge this paragraph with the conclusion chapter.
Author Response
Dear reviewer, all your suggestions have been implemented. The texts have been redrafted. Pictures exchanged. Improved all descriptions as recommended. Changes are visible in the text. The quality of the photos has also changed. New photos added. Unnecessary removed. More parameters of the welding process and more material parameters are given. A more in-depth description has been carried out. I will be grateful for positive acceptance of the text.

Reviewer 2 Report
In my opinion, this is a brief description of the experimental measurements carried out, the results of which are not analysed in detail in this paper. However, the article does not explain the scientific contribution of the achieved results.
Comments:
1. Missing text in line 168.
2. Check the quality of the figures.
3. Fig 1 - illegible description in Polish.
4. Fig 4 - appropriate to add scale.
5. Fig 9 - Where are the cracks that appear in lines 428, 429? Appropriate to mark a), b).
6. Fig 11 - replace - poor quality.
7. How do you explain the different displacement values at approximately the same loading forces shown in Table 3? (1752 kN / 38 mm and 1746 kN / 28 mm).
8. Lines 193/194 and 361 state, that a total of 20 experimental measurements were made, but the results are not given in detail.
Author Response

(The authors gave the same response as above.)

Reviewer 3 Report
This manuscript addresses issues with control and quality assurance of welded railway track joints using a series of destructive and nondestructive tests on the welded joints produced using the spark butt welding method. The methods utilized in this study include visual inspections (including inspections of the microstructure of the welded region), geometric tests, magnetic particle penetration tests, mechanical bend tests (that include loading to failure) and hardness tests. The applicability of each of these test methods is evaluated, including the limitations of each method. Overall, while the research methodology is sound, I believe this article is better suited for another journal such as MDPI Metals (more focused on the inspection techniques rather than characterization of the material itself). There are a few points that need to be addressed by the authors:
1. There are several grammatical errors in the manuscript (for instance, in the sentence "Due to security very much the high quality of the connections is important." in the Introduction section). An extensive editing of the English language is required.
2. Although technical conditions for welding are provided in Polish/European standards (PN-EN 14587-2), why are inspection techniques, whether destructive or non-destructive, not standardized for monitoring the structural health of welded railway track joints? Please elaborate.
3. This study uses spark butt welding method for fabricating the welded joints. The authors state that this method is more economical when compared to welding. What type of welding are the authors referring to? Is it arc welding?
4. Operational tests were carried out during 1 year of operation. Is this typical? Why were longer duration joints not tested? (5 years or 10 years of service). What is the typical inspection interval?
5. It appears that the geometric requirements stated in Tables 1 and 2 - are from Polish/ European standards (referenced in refs. 48 and 49). Please clarify since the references are in Polish.
6. Please provide scale bars in the images of the microstructure of the welds in Figure 4.
7. Please provide details of the images of the penetrant test in Figure 9. What is the magnification? How do we know there is a lack of surface defects? And what is the length scale of the surface defects that the penetrants can identify?
Author Response

(The authors gave the same response as above.)

Round 2
Reviewer 1 Report
I think the paper can be accepted in present form.
Reviewer 3 Report
The authors have addressed the reviewer comments satisfactorily.